# Mercury Phytotoxicity and Tolerance in Three Wild Plants during Germination and Seedling Development

**DOI:** 10.3390/plants11152046

**Published:** 2022-08-05

**Authors:** Carolina Kalinhoff, Norma-Thalia Calderón

**Affiliations:** Facultad de Ciencias Exactas y Naturales, Departamento de Ciencias Biológicas y Agropecuarias, Universidad Técnica Particular de Loja, San Cayetano Alto s/n, Loja 1101608, Ecuador

**Keywords:** Asteraceae, heavy metal, Malvaceae, radicle length, T50

## Abstract

By examining plant responses to heavy metal stress during the early stages of the life cycle, we can predict their tolerance and survival in polluted areas as well as their potential for bioremediation. The objective of our study was to evaluate the effect of exposure to mercury (Hg) on the germination and in vitro development of three plant species: *Bidens pilosa*, *Taraxacum officinale* (Asteraceae), and *Heliocarpus americanus* (Malvaceae). These are wild ecotypes adapted to local edaphoclimatic conditions in southern Ecuador, an area which has been historically affected by artisanal and small-scale gold mining (SSGM). For comparison, we additionally used a known Hg-tolerant plant, *Lactuca sativa* (Asteraceae). We tested biorelevant concentrations of Hg, equivalent to those occurring in soils affected by SSGM, i.e., up to 4.0 mg/L of Hg. The relative inhibitory effects of the treatments (0.6, 2.0, and 4.0 mg/L of Hg) on the germination percentage were most evident in *T. officinale*, followed by *B. pilosa*, while *L. sativa* and *H. americanus* were not affected. In terms of the time needed to reach 50% germination (T50), *B. pilosa* exposed to higher concentrations of Hg showed an increase in T50, while *H. americanus* showed a significant reduction compared to the control treatment. The reduction in radicle length at 4.0 mg/L Hg compared to the control was more evident in *L. sativa* (86%) than in *B. pilosa* (55.3%) and *H. americanus* (31.5%). We concluded that, in a scenario of Hg contamination in the evaluated concentration range, the grass *B. pilosa* and the tree *H. americanus* could have a higher probability of establishment and survival.

## 1. Introduction

Artisanal and small-scale gold mining (SSGM) provides income to more than 16 million people from economically depressed communities in more than 80 countries around the world [1,2]. Ten to fifteen million people living in developing countries extract gold by hand [3]. However, like most extractive anthropogenic activities, they generally have a negative and large-scale impact on natural ecosystems due to air, soil, and water pollution [4,5,6]. Although artisanal and SSGM provides economic opportunities for local people, it also causes negative impacts [7]. Historically, amalgamation with mercury, or leaching with sodium cyanide, or both were the most widely used process for gold recovery [8]. Mercury (Hg, or at times referred to as quicksilver) is a toxic, persistent, and mobile trace element with high volatility and low solubility, naturally present in the soil at low concentrations [9]. Mercury is the only metal that exists in a liquid aggregate state under standard conditions and evaporates on contact with the ambient air [10]. Atmospheric contamination by Hg is one of the most significant environmental problems in the modern world [11].

The magnitude and complexity of problems generated by Hg contamination demand the development of technologies adapted to specific environmental contexts. Phytoremediation is an environmentally friendly alternative for repairing heavy metal–polluted soils and water, through plant capacity for extracting heavy metals [12]. Although Hg tends severely to limit plant productivity even at low concentrations (between 1.0 and 1.5 mg/L), some tolerant plant species have been proposed for mercury phytoremediation, i.e., *Jatropha curcas*, *Erato polymnioides*, and *Cyrtomium macrophyllum* [1,13,14]. Nevertheless, Hg hyperaccumulators have not been found yet [15]. There is an urgent need for a better understanding of the response to Hg in widely distributed ecotypes, as well as in species native to each region.

Phytotoxicity tests are important to determine how different plants respond to environmental stress related to contamination [16,17]. These tests have been created in the context of contamination indicator methodologies, but their use as a preliminary tolerance test of new plants for Hg phytoremediation has not been fully exploited. Heavy metal toxicity is observed at all stages of a plant’s life cycle, but this effect is most pronounced during germination and radicle growth [18,19]. For this reason, it is possible that certain species that are tolerant during germination and seedling development can also be tolerant at later stages of their life cycle. A lower phytotoxic effect during germination or seedling growth might be a good predictor of successful establishment in contaminated areas [20].

The effect of Hg on the germination and growth of seedlings has allowed the recognition of new native plants with potential for phytoremediation around the world. In Mexico, native ecotypes of *Brassica napus* L. and *Celosia cristata* L. showed germination percentages between 67 and 70% in mine tailing with 1.0 mg/kg Hg, but only *B. napus* produced seedlings 15 days after germination. [21]. Three native Australian grasses used in restoration tolerated maximum inhibitory concentrations between 10 and 250 mg/kg of Hg, which affected their root elongation [22]. Five of six native populations of *Quercus ilex* subsp. *ballota* (Fagaceae) from old Hg mines and nearby areas in Spain showed high germination rates and good growth, up to 50 uM Hg [23].

Asteraceae and Malvaceae are species of diverse cosmopolitan plant families that possess wide adaptability and mechanisms that allow a higher tolerance to environmental stress, making them model organisms for phytoremediation [24,25,26]. Numerous species from Asteraceae and Malvaceae have a high potential to transform, immobilize, and degrade contaminants present in soil and water through phytoremediation processes, such as rhizofiltration, phytostabilization, phytoextraction, phytovolatilization, and phytotransformation [27,28,29]. In Ecuador, the Asteraceae and Malvaceae families are widely distributed, including in montane forests and montane shrublands affected by artisanal gold mining.

The scope of our research is to generate efficient phytoremediation strategies using native ecotypes in habitats that are affected by mining activity in southern Ecuador. We propose three species which are common in the region but have not been evaluated for their Hg tolerance: *Bidens pilosa*, *Taraxacum officinale* (Asteraceae), and *Heliocarpus americanus* (Malvaceae). *Bidens pilosa* can accumulate Pb in its roots [27] and *T. officinale* can also accumulate heavy metals such as Cd, Cr, Cu, Fe, Pb, and Zn in their leaves and roots [30]. *Heliocarpus americanus*, a tree native to Central America and distributed from southern Mexico to northern Argentina, is tolerant to Al [31]. This species is also considered a pioneer tree in forests, along riverbanks, and in degraded land [32]. Then, this study evaluated the seed germination and seedling growth of these plant species in Hg concentrations reported from artisanal and small-scale gold mining soils (up to 4.0 mg/L). *Lactuca sativa* was also included as a reference species tolerant to Hg.

## 2. Results

### 2.1. In Vitro Test

Exposure to Hg had a significant effect on *T. officinale* and *B. pilosa* (*p* ≤ 0.05; Figure 1). *Taraxacum officinale* showed a decrease in germination by 73.5% in the lowest concentration compared to the control, and no germination was found at higher concentrations of Hg (Figure 1). *Bidens pilosa* maintained similar germination percentages in the control, 0.6, and 2.0 mg/L Hg treatments; however, at 4.0 mg/L Hg, germination was reduced by almost 50% (Figure 1). In *L. sativa* and *H. americanus,* germination was not affected in the three treatments compared to control (Figure 1).

In *L. sativa*, the two intermediary treatments had a longer T50 compared to both the control and 4.0 mg/L Hg treatment. The T50 of *B. pilosa* was significantly longer at the highest Hg concentration compared to the other treatments. *Heliocarpus americanus* showed a significant reduction in T50 in all Hg treatments compared to the control (Figure 2).

### 2.2. Seedling Growth

Hypocotyl and radicle length decreased gradually with increasing Hg concentrations for all species (Table 1). At the highest concentration of Hg (4.0 mg/L), *L. sativa*, *B. pilosa,* and *H. americanus* had reduced radicle lengths by 86%, 55.3%, and 31.5%, respectively, compared to the control (Table 1). Hypocotyl lengths for these species had less pronounced reductions: 57.6%, 26.4%, and 28.6% for *L. sativa*, *B. pilosa*, and *H. americanus,* respectively. For *T. officinale*, few seedlings germinated in the 0.6 mg/L Hg treatment; these seedlings had 22.8% shorter hypocotyls but a similar radicle length compared to the control (Table 1).

In *L. sativa*, hypocotyl and radicle diameters were similar in all treatments, while in *B. pilosa* these two parameters were only reduced at 4.0 mg/L Hg. In *Heliocarpus americanus,* the radicle diameters did not significantly change; however, the hypocotyl diameter decreased by 20% at 4.0 mg/L Hg compared to the control (Table 2).

## 3. Discussion

Our results showed the relative inhibitory effects of Hg on germination percentage, in decreasing order: *T. officinale* > *B. pilosa* > *L. sativa* > *H. americanus*. The germination response to mercury of these species has not been previously reported (except for *L. saliva*), but Hg accumulation in adult plants has been reported for *T. officinale* and *B. pilosa*. A low proportion of Hg has been observed in *T. officinale* roots (0.00037 mg/kg) compared with the leaves (11.56 mg/kg) in adult plants [33]. Another investigation indicated the absence of correlation between soil and *T. officinale* tissue metal concentration [34], indicating the possible presence of repressive mechanisms in this species, or a high root susceptibility to this metal. The inhibitory mechanisms of seed germination and embryo development caused by Hg are practically unknown, but Hg replaces the -SH groups of proteins with S–Hg–S bridges, affecting amylase and protease activities crucial for seed germination [18]. Furthermore, it is possible that damage to the seed, physiological alterations, and the generation of reactive oxygen species (ROS) [35] impact *T. officinale* to a great extent during its germination.

The Hg inhibitory effect on *B. pilosa* germination at 4.0 mg/L indicates that its tolerance to this metal is low but not negligible compared to *L. sativa* and *H. americanus*. Tolerance to Hg in *B. pilosa* during germination has not been previously characterized, but moderate bioaccumulation has been reported in adult plants when exposed to other xenobiotics metals, such as Pb [36], Ni, Cr [37], and Cd [38], in addition to physiological differences between Cd accumulator ecotypes [39]. 

Regarding *L. sativa*, its tolerance during germination was higher than that of all the herbaceous plants evaluated in our study and similar to previous reports of germination with Hg. *L. sativa* shown decreased in root length by 17% at 10 mg/kg Hg, and only at very high concentration (100 mg/kg), hypocotyl and radicles formed aerenchyma [35]. In another study, it was reported that the radicle length of *L. sativa* seedlings is affected by mercury, and that toxicity is canceled in the presence of arbuscular mycorrhizal fungi [40]. In addition, *L. sativa* was the most resistant species to Hg (II) [41] in a multitrophic battery test. This is a toxicological test using species known to be especially sensitive to a wide range of pollutants, representative of a range of trophic levels, and with standard and reproducible responses to facilitate inter-laboratory comparisons [42]. 

*H. americanus* did not show germination inhibition in the 4.0 mg/L Hg treatment, indicating a higher tolerance. Other tree and shrub species such as *Gossypium hirsutum* (cotton) were able to maintain 80% germination at 200 mg/kg Hg [43], and *Quercus ilex* did not have decreased germination at 10 mg/kg Hg, but its root length was reduced at 1.0 mg/kg Hg [44]. On the other hand, the neotropical tree *Plathymenia reticulate* (Fabaceae), from the Brazilian Cerrado reduced its germination by 80% when exposed to 10^−3^ M HgCl_2_ for 72 h [45]. *Heliocarpus americanus* can likely tolerate more than 4.0 mg/L Hg since its germination was unchanged up to that concentration, and its median germination time was reduced. 

Median germination time (T50) is one of the most important parameters in the evaluation of toxicity of heavy metals in plants [46]. Our results of higher T50 in *B. pilosa* and *L. sativa* with increasing concentrations of Hg have been reported when seeds are exposed to highly toxic metals such as Hg and Cd [20]. Only *H. americanus* had a decreased T50 in all Hg concentrations compared to the control. A lower T50 value could be a result of a scarifying effect on the seeds [47] or the activation of cellular enzymatic and non-enzymatic antioxidants against reactive oxygen species—ROS [48,49].

The measurement of parameters in seedling development when exposed to heavy metals gives a baseline knowledge of their tolerance [20] and facilitates the pre-selection of species for future complex and expensive experiments. In our study, radicle lengths were more sensitive to Hg than germination percentage. This trend coincides with a higher toxicity of Hg and Pb during seedling growth compared to germination in *Arabidopsis thaliana* [50]. Despite having high germination percentages, *L. sativa* seedlings were the most affected by Hg in our study compared to *B. pilosa* and *H. americanus*. The reduction in radicle length at 4.0 mg/L compared to the control was more evident in *L. sativa* (86%) than in *B. pilosa* (55.3%) and *H. americanus* (31.5%). Similar results of root cell damage in *L. sativa* were reported by [35] but at higher concentrations of Hg (10 mg/kg). In other species such as *Vigna radiata* L., inhibition of seedling growth at low concentrations of Hg (1.0 and 1.5 mg/L) has been reported [51].

In Ecuador, it is urgent to characterize the high plant diversity historically associated with soils affected by mining to propose effective phytoremediation strategies for different ecosystems. The ability of seeds to germinate in the presence of heavy metals and metalloids could be the main limitation in ecosystem rehabilitation [52]. Currently, trees are being used in phytosanitary detection technologies by assessing tree core sampling, phytoextraction and phytostabilization of Cd, Cu, Ni, and Zn [53]. *Heliocarpus americanus* is a short-lived, fast-growing pioneer tree which improves its productivity with the association of native arbuscular mycorrhizal fungi [54]. According to our results, it is possible that *H. americanus* germinates in soils contaminated with Hg in southern Ecuador. This tree can be considered a promising species for the phytoremediation of Hg in mining areas. In this context, possible alternatives are to evaluate their response to Hg in later stages of development, including consideration of the substrates where populations will ultimately establish.

## 4. Materials and Methods

### 4.1. Collection, Sterilization, and Storage of Seeds

Fruits of *H. americanus*, *B. pilosa* and *T. officinale* were collected in Zamora Chinchipe Province, southern Ecuador. Commercial *L. sativa* seeds (Great Lakes 118) were also used since this is recognized as an Hg tolerant plant [24,55]. A total of 2000 seeds of each species were immersed in 70% ethanol for 1 min, then in a 1% sodium hypochlorite solution for 10 min, and rinsed three times with deionized water for 10 min. Seeds were afterwards air-dried and stored for 30 days in paper bags before the start of the experiment.

### 4.2. In Vitro Test

Hg treatments were prepared from a stock solution of 4.0 mg/L of HgCl_2_. To verify the non-lethal range of Hg on tested seeds, a preliminary assay was carried out with 0.0, 0.6, and 2.0 mg/L of Hg (three replicates with 10 seeds for each treatment). Percentage of germination in *B. pilosa* and *L. sativa* were not affected at 0.6 and 2.0 mg/L Hg compared to the control, whereas germination of *H. americanus* increased at 2.0 mg/L Hg (Table A1). Conversely, the germination of *T. officinale* was reduced in the lowest concentration of Hg, showing low tolerance to this heavy metal. According to these preliminary results, an additional Hg concentration (4.0 mg/L) was incorporated into the final experiment. 

Subsequently, the experimental setup was established using concentrations of 0.0 mg/L (control), 0.6 mg/L, 2.0 mg/L, and 4.0 mg/L Hg. Seeds were placed in Petri dishes on two layers of Whatman N^o^ 1 filter paper. In each Petri dish, 25 mL of sterile distilled water (control) or the respective Hg solution was added. The parafilm-sealed Petri dishes were kept in a growth chamber for 12 days with 16 h of light and 8 h of darkness at 23.5 °C and 65% humidity. The number of germinated seeds was quantified using a stereo microscope every 24 h.

### 4.3. Germination Response Variables

Germination was calculated using the formula: Germination (%) = (Number of germinated seeds/Total number of seeds) × 100. 

Median germination time (T50), defined as the time until 50% of the seeds reach final germination divided by the duration of the experiment, was calculated using the formula:T_50_ = [N/2 − n_i_] × (t_i_ − t_j_)/(n_i_ − n_j_)
where N is the final number of germinated seeds, n_i_ and n_j_ are the cumulative number of germinated seeds in consecutive counts, and t_i_ and t_j_ are time, measured in hours [56]. 

### 4.4. Growth Response Variables

After 12 days of exposure to Hg, hypocotyl and radicle length and diameter were measured with a digital caliper measuring tool, CACHOR 0–6 Vernier. Seedlings were washed with sterile distilled water and air-dried for 5 min on absorbent paper before measurements.

### 4.5. Experimental Design and Statistical Analysis

The experimental design was completely randomized, with three replicates containing 20 seeds per petri dish for each treatment. Germination and T50 were measured 7 days after Hg exposure, and the diameter and length of the hypocotyl and radicle were measured in the same sets of germinated seeds 12 days after Hg exposure. The data were analyzed using a one-way ANOVA test for each species, performed in RStudio^®^ with the ‘Anova’ function from the ‘Car’ package in R [57]. A Tukey’s post hoc test was applied to find out which treatments were statistically different. The significance level was set at α = 0.05. Graphs were made in Sigmaplot 11.0 package (Systat Software, Inc., San Jose, CA, USA).

## 5. Conclusions

Mercury pollution is an emerging global threat that requires sustainable responses for its management, especially in regions with a high practice of artisanal and SSGM. Germination and seedling development screening is a rapid and practical form to generate information related to the response of many plant species to heavy metal exposure. The results of this experiment showed that *B. pilosa* and *H. americanus* are Hg tolerant and potentially an alternative for the phytoremediation of regional polluted soils. It is possible to preselect a higher number of tolerant native ecotypes by measuring germination and radicle length to use the most suitable species in future complex and expensive tests of tolerance to heavy metals. *Bidens pilosa* could be useful for the phytoremediation of Hg concentrations less than 2.0 ppm, while *H. americanus* could tolerate higher concentrations (<4 mg/L of Hg), but more studies are needed to characterize these species. We recommend further tests be carried out throughout their distribution specifically in polluted areas, including soil characteristics such as texture, pH, percentage of organic matter, temperature, and microorganisms. Particularly, the tree *H. americanus* could be a good candidate for assisted phytoremediation due to its rapid growth, high mycorrhizal response, tolerance to different edaphoclimatic conditions, and wide distribution in Central and South America. 

## Figures and Tables

**Figure 1 plants-11-02046-f001:**
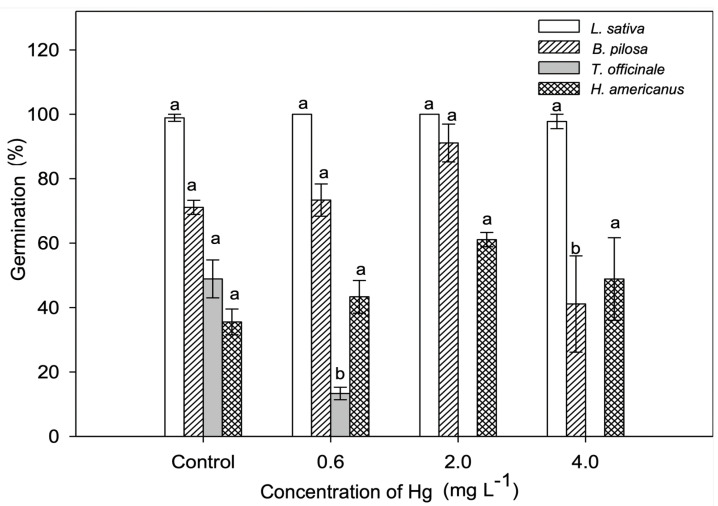
Germination percentage in *Lactuca sativa*, *Bidens pilosa*, *Taraxacum officinale,* and *Heliocarpus americanus* seeds exposed to increasing Hg concentrations (0.0 mg/L−control, 0.6 mg/L, 2.0 mg/L, and 4.0 mg/L). Values are expressed as means and standard errors are shown. Bars followed by different letters indicate differences between treatments within the same species (*p* ≤ 0.05) using the Tukey test.

**Figure 2 plants-11-02046-f002:**
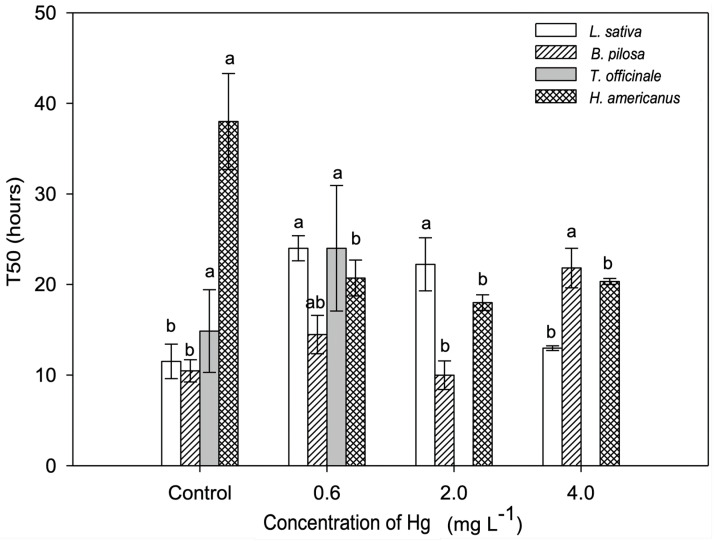
Median germination time (T50) in *Lactuca sativa*, *Bidens pilosa*, *Taraxacum officinale*, and *Heliocarpus americanus* seeds exposed to increasing Hg concentrations (0.0 mg/L−control, 0.6 mg/L, 2.0 mg/L, and 4.0 mg/L). Values are expressed as means and standard errors are shown. Bars followed by different letters indicate differences between treatments within the same species (*p* ≤ 0.05) using the Tukey test.

**Table 1 plants-11-02046-t001:** Hypocotyl and radicle length in *Lactuca sativa*, *Bidens pilosa*, *Taraxacum officinale,* and *Heliocarpus americanus* after 12 days of exposure to increasing Hg concentrations (0.0 mg/L−control, 0.6 mg/L, 2.0 mg/L, and 4.0 mg/L). Means are shown with standard deviations in parentheses. Different letters indicate differences between treatments within the same species (*p* ≤ 0.05) using the Tukey test.

Species	Treatment	Hypocotyl Length (mm)	Radicle Length (mm)
*L. sativa*	Control	38.9 (3.60) a	28.0 (5.35) a
	0.6	31.0 (2.01) b	13.2 (2.55) b
	2.0	16.6 (2.07) c	7.2 (2.33) c
	4.0	16.5 (2.18) c	3.7 (1.33) c
*B. pilosa*	Control	36.4 (11.04) a	15.0 (7.35) a
	0.6	37.7 (3.73) a	17.3 (3.01) a
	2.0	31.8 (3.28) a	10.9 (3.73) a
	4.0	26.8 (6.74) b	6.7 (4.82) b
*T. officinale*	Control	18.4 (2.08) a	5.4 (1.23) a
	0.6	14.2 (3.17) b	5.4 (1.27) a
	2.0	-	-
	4.0	-	-
*H. americanus*	Control	26.6 (2.95) a	9.2 (1.76) a
	0.6	26.6 (2.39) a	8.4 (1.25) a
	2.0	19.5 (4.04) b	6.2 (1.05) b
	4.0	19.0 (3.21) b	6.3 (0.94) b

**Table 2 plants-11-02046-t002:** Hypocotyl and radicle diameter in *Lactuca sativa*, *Bidens pilosa*, *Taraxacum officinale,* and *Heliocarpus americanus* after 12 days of exposure to increasing Hg concentrations (0.0 mg/L−control, 0.6 mg/L, 2.0 mg/L, and 4.0 mg/L). Means are shown with standard deviations in parentheses. Different letters indicate differences between treatments within the same species (*p* ≤ 0.05) using the Tukey test.

Species	Treatment	Hypocotyl Diameter (mm)	Radicle Diameter (mm)
*L. sativa*	Control	0.21 (0.03) a	0.09 (0.02) a
	0.6	0.17 (0.02) b	0.09 (0.01) a
	2.0	0.20 (0.02) a	0.09 (0.01) a
	4.0	0.22 (0.02) a	0.10 (0.02) a
*B. pilosa*	Control	0.16 (0.05) a	0.04 (0.02) a
	0.6	0.19 (0.04) a	0.04 (0.02) a
	2.0	0.18 (0.03) a	0.06 (0.01) a
	4.0	2.43 (3.84) b	1.07 (1.69) b
*T. officinale*	Control	0.18 (0.03) a	0.08 (0.02) a
	0.6	0.17 (0.03) a	0.06 (0.02) a
	2.0	-	-
	4.0	-	-
*H. americanus*	Control	0.20 (0.03) a	0.07 (0.02) a
	0.6	0.17 (0.03) b	0.06 (0.01) a
	2.0	0.20 (0.02) a	0.07 (0.01) a
	4.0	0.16 (0.03) c	0.06 (0.01) a

## Data Availability

Not applicable.

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
