# Peer review of "Mercury Phytotoxicity and Tolerance in Three Wild Plants during Germination and Seedling Development"

_plants, 2022, doi:10.3390/plants11152046_

Round 1

Reviewer 1 Report

Please include some latest research findings, updated reviews in introduction and discussion part related to the topic.

Many sentences do not have the correct punctuation and it is difficult to read the text.

English should be improved; grammar need for enhancement in many sentences and paragraphs.

The conclusion you have provided is quite brief and provide sufficient feedback on the main objectives of your study.

Author Response

Manuscript ID: plants-1836122

Title: "Mercury Phytotoxicity and Tolerance in Three Wild Plants During Germination and Seedling Development"

Journal: Plants

July 28Th, 2022

Dear Editor:
First, the authors wish to thank the reviewer and the editor for their suggestions and comments, which have contributed to improve significantly the quality of this manuscript.

In the following section, we provide a point-by-point response to the reviewer 1 comments.

Yours sincerely,

Carolina Kalinhoff.

The authors thank reviewer 1 for his important opinions and suggestions.

  1. Please include some latest research findings, updated reviews in introduction and discussion part related to the topic. Response: The scarce information about the species evaluated limited in some degree the comparability of our results. But, following reviewer 1's suggestion, we included 4 new references related to another species evaluated under similar conditions. These new references can be seen in an additional background paragraph in the introduction (lines 64-71). In the discussion, we had already incorporated updated references. In this opportunity, we add another article of germination response of a tree species (Lines 179-180).
  2. Many sentences do not have the correct punctuation and it is difficult to read the text. grammar need for enhancement in many sentences and paragraphs. Response: Long sentences were shortened and punctuation and grammatical errors corrected sentence by sentence.
  1. The conclusion you have provided is quite brief and provide sufficient feedback on the main objectives of your study. Response: Considering that reviewer 1 tells us that the conclusion provides sufficient feedback on the main objectives of our study, we kept it the same and only checked punctuation and grammar.

Reviewer 2 Report

I am satisfied with the changes made authors.

Author Response

Manuscript ID: plants-1836122

Title: "Mercury Phytotoxicity and Tolerance in Three Wild Plants During Germination and Seedling Development"

Journal: Plants

July 28Th, 2022

Dear Editor:
First, the authors wish to thank the reviewer and the editor for their suggestions and comments, which have contributed to improve significantly the quality of this manuscript.

In the following section, we provide a response to the reviewer 2 comments.

Yours sincerely,

Carolina Kalinhoff.

I am satisfied with the changes made authors. Response: We value reviewer 2's appreciation of all improvements made to the manuscript.

Reviewer 3 Report

The manuscript was improved based on the reviewers' requests. Missing information was added, discussions, material, and conclusions were enhanced as suggested.

In my opinion, the work could be accepted in the current form.

Best regards.

Author Response

Manuscript ID: plants-1836122

Title: "Mercury Phytotoxicity and Tolerance in Three Wild Plants During Germination and Seedling Development"

Journal: Plants

July 28Th, 2022

Dear Editor:
First, the authors wish to thank the reviewer and the editor for their suggestions and comments, which have contributed to improve significantly the quality of this manuscript.

In the following section, we provide a response to the reviewer 3 comments.

Yours sincerely,

Carolina Kalinhoff.

Reviewer 3 comment:

The manuscript was improved based on the reviewers' requests. Missing information was added, discussions, material, and conclusions were enhanced as suggested.

In my opinion, the work could be accepted in the current form.

Best regards.

Response: We are deeply grateful to reviewer 3 for approving the changes made to the manuscript. We also made minor spelling corrections indicated by reviewer 3.

This manuscript is a resubmission of an earlier submission. The following is a list of the peer review reports and author responses from that submission.

Round 1

Reviewer 1 Report

-Major shortcomings include the poor bibliometric analysis (data needs to be drawn from a major database such as Scopus or Web of Science, and a wider range of results should be displayed); and the lack of a subsection describing the

-The article is not well structured and not all references are adequately cited. Moreover, authors have not put a lot of afford in manuscript editing..

-All figures need to enhancement the resolution and legends not well described

-The English language is of the poor quality and needs to be substantially improved.

Reviewer 2 Report

After careful reading of the ms I found it suitable for publication, since the authors used suitable and standard methods and analysis. This is a good work. The abstract abides by all the editing instructions and presents the objectives of the study. The introduction is supported by well selected bibliographic data. Also discussion is rich, since many important scientific references have been taken into consideration. I suggest to publish the ms after minor revision, since I suggest to improve the ms as follows:

References

p. 4 and 5, line 411-492: correct references numbers

p. 5, line 484: correct year of publication of reference 2009

Reviewer 3 Report

The authors investigated the Hg tolerance and the potencial for bioaccumulation of three plant species which are widely distributed in Ecuador (Bidens pilosa, Taraxacum officinale, and Heliocarpus americanus). It was evaluated the effect of three dose of Hg (0.6, 2.0 and 4.0 mg L-1) on the germination percentage and seedling development (hypocotyl and radicle length, and diameter). Lactuca sativa was included as a reference spe cies tolerant to Hg.

On the basis of results, the authors identified  B. pilosa and H. americanus as the species with a higher probability  of establishment and survival.

In my opinion the work is interesting, clear, and well written, so it could be accepted in the current form.

Best regards.